# Deep Learning-Based Multi-Class Segmentation of the Paranasal Sinuses of Sinusitis Patients Based on Computed Tomographic Images

**DOI:** 10.3390/s24061933

**Published:** 2024-03-18

**Authors:** Jongwook Whangbo, Juhui Lee, Young Jae Kim, Seon Tae Kim, Kwang Gi Kim

**Affiliations:** 1Department of Computer Science, Wesleyan University, Middletown, CT 06459, USA; jwhangbo@wesleyan.edu; 2Medical Devices R&D Center, Gachon University Gil Medical Center, Incheon 21565, Republic of Korea; juhui05134@gmail.com (J.L.); youngjae@gachon.ac.kr (Y.J.K.); 3Department of Health Sciences and Technology, Gachon Advanced Institute for Health & Sciences and Technology (GAIHST), Gachon University, Incheon 21565, Republic of Korea; 4Department of Otolaryngology-Head and Neck Surgery, Gachon University Gil Hospital, Incheon 21565, Republic of Korea; kst2383@gilhospital.com; 5Department of Biomedical Engineering, College of IT Convergence, Gachon University, Seongnam-si 13120, Republic of Korea

**Keywords:** paranasal sinuses, chronic sinusitis, Convolutional Neural Network (CNN), multiclass segmentation

## Abstract

Accurate paranasal sinus segmentation is essential for reducing surgical complications through surgical guidance systems. This study introduces a multiclass Convolutional Neural Network (CNN) segmentation model by comparing four 3D U-Net variations—normal, residual, dense, and residual-dense. Data normalization and training were conducted on a 40-patient test set (20 normal, 20 abnormal) using 5-fold cross-validation. The normal 3D U-Net demonstrated superior performance with an F1 score of 84.29% on the normal test set and 79.32% on the abnormal set, exhibiting higher true positive rates for the sphenoid and maxillary sinus in both sets. Despite effective segmentation in clear sinuses, limitations were observed in mucosal inflammation. Nevertheless, the algorithm’s enhanced segmentation of abnormal sinuses suggests potential clinical applications, with ongoing refinements expected for broader utility.

## 1. Introduction

In 1994, around 200,000 sinus surgeries were conducted in the United States [1]. By 1996, 12 percent of Americans under the age of 45 reported symptoms indicative of chronic sinusitis [2]. This widespread condition imposes a substantial societal burden, manifesting in frequent office visits, absenteeism from work, and missed school days [1]. When medicinal treatments fail to alleviate the condition, patients are often referred for sinus surgery. Many physicians refer to Computed Tomography (CT) scans when evaluating patients referred for sinus surgery [1,3]. Radiologists report anatomic variants, that can affect operative techniques, and critical variants, that can complicate surgery [4]. Identification of these anatomical variants affords the opportunity to avoid surgical complications [5]. Segmentation data can be used for the diagnosis, surgical planning, or workspace definition of robot-assisted systems. However, manual and semiautomatic segmentation of the paranasal sinuses has been evaluated as impractical in clinical settings because of the amount of time required for both systems [6,7]. The application of machine learning in this process warrants attention due to its potential to substantially mitigate the time and labor costs associated with manual segmentation. Ultimately, this holds promise for making the segmentation process feasible and practical in clinical settings.

Artificial intelligence is gaining popularity in the medical imaging field for developing models that produce human-interpretable results [8,9,10]. Because of the clustered arrangement of regions, including the frontal, ethmoid, and sphenoid sinuses, developing models that can produce practical results for the paranasal sinuses is an ongoing challenge. Two published studies focused on processing cone-beam computed tomography images to achieve segmentation of the maxillary sinus. In 2022, Choi et al. [11] trained a U-Net model to segment maxillary sinuses. The segmented results were refined using post-processing techniques to isolate and remove disconnected false positives. The trained model made predictions with a Dice similarity coefficient (DSC) value of 0.90 ± 0.19 before post-processing and 0.90 ± 0.19 after post-processing. Morgan et al. [12] trained two U-Net models to segment the maxillary sinus. The first model suggested crop boxes in the original image of the maxillary sinus, which were used to train the second part of the model to produce high-resolution segmentation results. The final segmentation results achieved a DSC score of 0.98 for the first model and 0.99 for the second model. Kuo et al. [13] proposed a 6-class segmentation model that segmented four different areas of the paranasal sinuses, treating the ethmoid sinus as two different areas: the anterior and posterior ethmoid sinus. A secondary model was trained to generate pseudo-labels on the unlabeled datasets. The model used in this study was an adaptation of the U-Net model [14] with the addition of depth-wise separable convolution, squeeze-and-excitation networks, and residual connections. The model was able to make predictions with a DSC value of 0.90. The approaches proposed by Choi et al. [11] and Morgan et al. [12] exhibited performance adequate for clinical applications. However, the aim of both studies was limited to the binary segmentation of the maxillary sinus.

We proposed a 5-class segmentation model for the four regions of the paranasal sinus: frontal sinus, ethmoid sinus, sphenoid sinus, and maxillary sinus. Training and validation were conducted on clinical-level CT scans sourced from patients exhibiting high degrees of genetic and biological variations. The objective was to develop a model capable of generating clinical data with sufficient accuracy to be practically applicable in clinical settings.

## 2. Materials and Methods

This study was approved by the Institutional Review Board (IRB) of Gachon University Gil Medical Center (GAIRB2022-182) and was conducted in accordance with the relevant guidelines and ethical regulations.

A total of 39,605 paranasal CT scans were collected from 201 patients with varying degrees of chronic sinusitis, including 3821 images from 20 patients without sinusitis. A total of 40 datasets were randomly selected as the hold-out test set, with 20 datasets originating from the patient group without sinusitis. These subsets were then labeled as “normal” and “abnormal” to reflect the respective patient group characteristics. Training was performed on the remaining 161 datasets with 5-fold cross validation, where 128 datasets were used for training and 33 for validation. In summary, the dataset was divided into sets comprising 128 patients for training, 33 patients for validation, and 40 patients for testing. Demographic information of the participating patients is summarized in Table 1.

Data collection and storage were performed using Excel (version 16.83, Microsoft, Redmond, WA, USA) and statistical analyses were performed using MedCalc (version 22, MedCalc Software Ltd., Ostend, Belgium). Training was performed on an Ubuntu server (version 20.04.6 LTS) with four Nvidia A100 80Gb GPUs (NVIDIA, Santa Clara, CA, USA), an AMD EPYC 7452 32-Core Processor (AMD, Santa Clara, CA, USA), and 1,031,900 Mb of RAM. The following libraries were used for training: Python (version 3.7), TensorFlow (version 2.6.0), and Keras (version 2.6.0).

Using the collected sinus data, we meticulously curated a ground truth dataset by labeling the sinus region for each patient. The oversight and guidance of two experienced otorhinolaryngologists was integral to this process, ensuring the utmost quality and accuracy of the dataset. The final ground truth data were congregated through a consensus between the two physicians. The ground truth was labeled along the axial, sagittal, and coronal axes, as visually depicted in Figure 1. The volumetric reconstruction (Figure 1d) presents the data in its authentic form, providing insight into how it is inputted into the deep learning model. The axial view (Figure 1a) shows the maxillary and sphenoid sinuses beneath the ethmoid sinuses. The sagittal view (Figure 1b) shows the left maxillary sinus and part of the sphenoid sinus. The coronal view (Figure 1c) shows the frontal and maxillary sinuses surrounding the ocular area.

To facilitate the extraction of features within the CT scans, the datasets underwent several enhancements (Figure 2A), including window setting adjustments, isotropic voxel reconstruction, contrast-limited adaptive histogram equalization (CLAHE), and region of interest (ROI) cropping. The preprocessed images were used to train the segmentation model (Figure 2B) to produce segmentation results (Figure 2C). The overall training process is presented in Figure 2. A bone window with a width of 2,000 and a level of 0 was set and converted into 8-bit encoding. This setting has been established as the imaging technique of choice for examining patients before functional endoscopic sinus surgery [15,16].

Depending on the acquisition process, CT images can have varying slice thickness and pixel spacing within the protocol range [17,18]. The acquired images exhibited a consistent 1 mm slice thickness but varying pixel spacings, resulting in the disproportionate volumetric ratio of planar CT images. To eliminate unwanted ratio variations among the dataset, an isotropic voxel reconstruction algorithm was applied across the dataset to equalize the slice thickness to pixel spacing ratio. The ratio of slice thickness to pixel spacing was calculated to downsample the images accordingly using cubic spline interpolation [19] such that the volumetry of the resized images matched real proportions.

Adaptive histogram algorithms are commonly used in medical imaging to create images with equal intensity levels, thereby generating an image with an increased dynamic range, leading to an increase in contrast [20,21]. CLAHE [22,23] was employed in this study to restrict amplification and prevent overamplification of noise in areas with relatively homogeneous contrast.

To equalize the image dimensions for training, a cropping algorithm was used to crop images based on the region of interest. To guarantee the comprehensive inclusion of the region of interest, specific dimensions were set, with a target depth of 192, a height of 128, and a width of 128. The dimensions were chosen based on an analysis of the ground truth data in the entire dataset. The algorithm used in the analysis calculated the 3-dimensional coordinates of the edges for the largest ground truth data. As the voxel reconstruction algorithm resized the CT scans in accordance with the actual proportions of the paranasal sinuses, a greater amount of ground truth data became available along the depth axis.

The U-Net architecture is commonly used for medical image segmentation models because of its reliable performance on medical images [24,25,26]. Furthermore, its utilization of depth-wise 3D convolution operations allows for the simultaneous extraction of features along the 3 axes: axial, sagittal, and coronal. Three variants of the 3D U-Net architecture, each deeper than the last, were trained and compared: 3D U-Net with residual connections [27], 3D U-Net with dense blocks [28], and 3D U-Net with dense blocks and residual connections [29]. The 3D U-Net architecture, which served as the basis for constructing our model, is presented in Figure 3.

The 3D U-Net used in this study comprised 18 convolutional layers with 5,644,981 trainable parameters. The residual 3D U-Net comprised 63 convolutional layers and 2,350,989 trainable parameters. The dense 3D U-Net comprised 28 convolutional layers and 10,960,437 trainable parameters. The residual dense 3D U-Net comprised 34 convolutional layers and 47,078,117 trainable parameters. A summary of the parameter and layer counts for each model is provided in Table 2, along with the kernel-wise feature map details summarized in Table 3. All models were trained on the same hyperparameters. The Adam [30] optimizer was used with an initial learning rate of 0.0001. Categorical cross-entropy loss was used to monitor validation loss, and accuracy was used as the evaluation metric. Learning rates on plateaus, early stoppers, and model checkpoints were used to prevent issues such as overfitting and plateauing. The tolerance for learning rate reduction was configured to 20 epochs, while the early stopper tolerance was set at 30 epochs.

## 3. Results

Each model was tested against the hold-out test set to generate segmentation results. The segmentation results were evaluated using the following five performance metrics: intersection over union (IoU), accuracy, recall, precision, and F1 score. The results are expressed as the mean ± 95% confidence interval, with statistical significance set at *p* < 0.05.

The segmentation results from the normal test set were evaluated using the performance metrics and summarized in Table 4. Overall, the models were able to make predictions with an F1 score in the range of 0.843–0.785, of which the 3D U-Net model achieved the highest F1 score with a value of 0.843. Conversely, the residual 3D U-Net model recorded the lowest F1 score, standing at 0.785.

The segmentation results from the abnormal test set are summarized in Table 5. In the abnormal test set, the segmentation results were evaluated to record a lower overall F1 score in the range of 0.793–0.740. The 3D U-Net model made predictions with the highest F1 score of 0.793, whereas the predictions made by the residual-dense 3D U-Net model recorded the lowest F1 score of 0.741.

A comparative plot of IoU values across the models in the normal and abnormal test set is presented in Figure 4. The average IoU difference across the models was 0.082 ± 0.034 (mean ± 95% confidence interval). Paired *t*-tests of the IoU across the models showed statistically insignificant differences in IoU values between the four models (*p* < 0.05). The average F1 score difference, encompassing both test sets, between the 3D U-Net and the other three models were as follows: 0.067 ± 0.016 for the residual model, 0.069 ± 0.028 for the dense model, and 0.082 ± 0.037 for the residual-dense 3D U-Net. Paired *t*-tests of the F1 scores between the models showed statistically insignificant F1 score variation across the models (*p* < 0.05). The average differences in F1 scores between the two test sets (normal and abnormal) were as follows: 0.170 ± 0.067 for the 3D U-Net, 0.188 ± 0.064 for the residual 3D U-Net, 0.206 ± 0.072 for the dense 3D U-Net, and 0.257 ± 0.099 for the residual-dense 3D U-Net. Statistical analysis using paired *t*-tests showed a statistically significant difference in the F1 scores between the normal and abnormal test sets (*p* > 0.05).

Visual overviews of the segmentation results for the normal and abnormal test sets are shown in Figure 5 and Figure 6, respectively. The figures show the segmentation results for the ethmoid sinus, maxillary sinus, and sphenoid sinus; each area is color-coded for better visual representation. The images were chosen randomly from the fold with the best mIoU score. Each row represents predictions from different models. From left to right, the three columns represent the ground truth, prediction, and overlay comparison of the ground truth and prediction.

Normalized true positive (TP) distribution per class as a heatmap for the 3D U-Net is shown in Figure 6. For the normal dataset, the sphenoid sinus showed the highest TP rate of 0.95, whereas the ethmoid sinus showed the lowest at 0.82. For the abnormal dataset, the sphenoid sinus reported the highest TP rate at 0.88, and the lowest for the frontal sinus at 0.67.

## 4. Discussion

In this study, a 3D segmentation model for the four areas of the paranasal sinus based on CT images was developed and evaluated. Four models based on the 3D U-Net were trained and evaluated on a hold-out test set of 40 datasets, comprising 20 datasets from patients without sinusitis and 20 datasets from patients with sinusitis. Prediction results were further validated using 5-fold cross validation. In the normal test set, the models showed performances in the range of 0.843–0.785 with an average F1 score of 0.805. In the abnormal test set, the models performed in the range of 0.793–0.740 with an average F1 score of 0.755. In both test sets, the base 3D U-Net was able to make predictions with the highest F1 score of 0.843, and 0.793, respectively, in the normal test set and the abnormal test set. Statistical analysis of performance metrics was performed across the four models between normal and abnormal test sets with statistical significance set at *p* = 0.05. Performance metrics across the models exhibited statistically insignificant variations. However, mucosal inflammation had a greater impact on the performance metrics across the models.

The method proposed by Choi et al. [11] reported an F1 score of 0.972 in normal sinuses and 0.912 in sinuses with mucosal inflammation. Morgan et al. [12] reported an F1 score of 0.984 and 0.996, respectively, for normal and abnormal sinuses. Note that these studies were limited to binary segmentation of the maxillary sinus, manifesting in the higher F1 score. The study by Kuo et al. [13] trained multiple models with the aim of multi-class segmentation of the sinus, in which the U-Net model reported an average F1 score of 0.896. This is within 6.2% of the highest performing model in our study, the base 3D U-Net.

We performed a thorough analysis of prediction accuracy for the 3D U-Net model across the four main sinus regions, focusing on true positive rates. The outcomes underscored notable limitations in the precise prediction of the frontal and ethmoid sinus regions. The abnormal test set showed lower prediction metrics, overall, in comparison to the normal test set. The frontal and ethmoid sinuses showed particularly lower TP rates in the abnormal test set, at 0.67 and 0.75, respectively. The frontal and ethmoid sinuses are anatomically adjacent structures, and both have smaller volumes than the sphenoid and maxillary sinuses [31]. In sinus cavities with mucosal inflammation, the cavities of the ethmoid and frontal sinuses had much less pronounced features compared to other areas of the paranasal sinuses. This limitation is evident in Figure 7 of the right ethmoid sinus, where the contrast between the sinus bone and cavity appears less pronounced compared to the left ethmoid sinus.

Despite the substantial size of the dataset collected for this study, the clinical nature of the CT scans led to an uneven distribution of data between patients with sinusitis and those without the condition. Moreover, training data was obtained solely from a single institution, suggesting the possibility that the trained models could exhibit limited generalization capabilities on external datasets. A comprehensive follow-up study should encompass a well-balanced dataset, including an equal distribution of data from patients with sinusitis and those without the condition. It would be advantageous to source this data from multiple institutes to enable internal and external validations.

Accurate segmentation of the paranasal sinuses is crucial for the preoperative evaluation of patients undergoing sinus surgery. To this end, this study aimed to evaluate the segmentation efficacy in patients with mucosal inflammation. While limitations do exist in the segmentation of paranasal sinuses with mucosal inflammation, the proposed method exhibited promising results. With minor refinements, our segmentation model has the potential to enhance surgical accuracy when integrated into guidance systems. Such integration can aid surgeons in avoiding healthy mucosal tissue, thereby reducing the risk of complications.

## Figures and Tables

**Figure 1 sensors-24-01933-f001:**
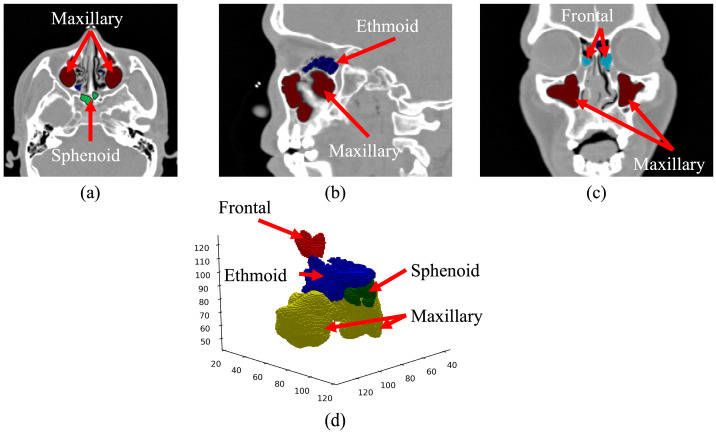
CT image of the paranasal sinuses with ground truth data overlayed. (**a**) Axial view, (**b**) sagittal view, (**c**) coronal view, and (**d**) volumetric reconstruction.

**Figure 2 sensors-24-01933-f002:**
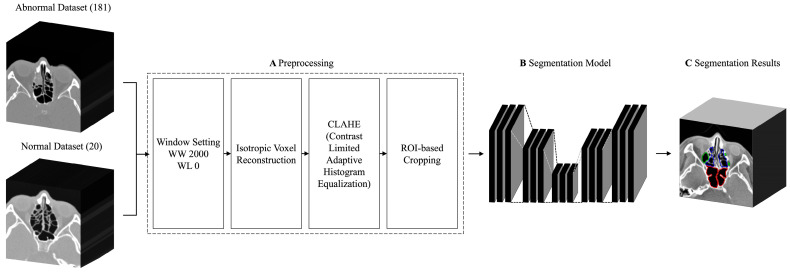
Flowchart of the multiclass sinus segmentation training process.

**Figure 3 sensors-24-01933-f003:**
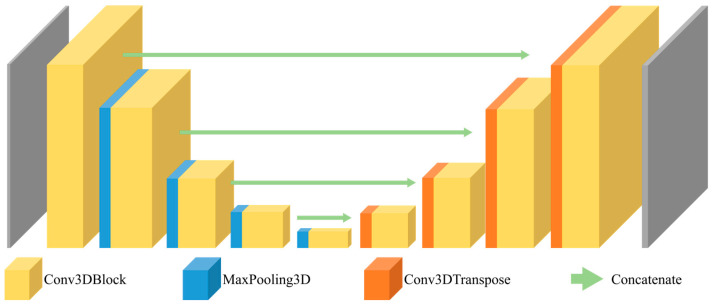
Architecture of the 3D U-Net.

**Figure 4 sensors-24-01933-f004:**
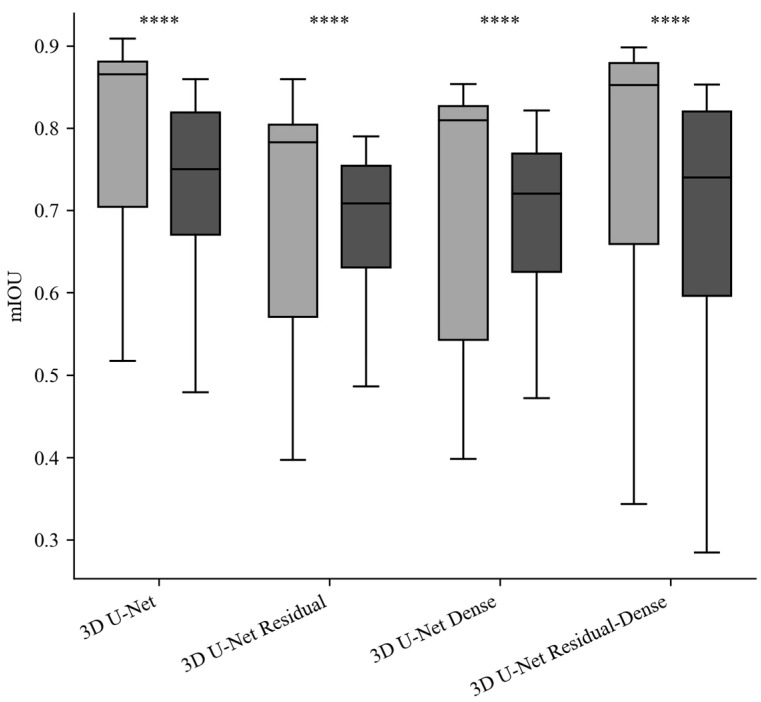
mIoU comparison of each model in the normal and abnormal test set (gray: normal dataset results, black: abnormal dataset results, ****: statistical significance (*p* < 0.05)).

**Figure 5 sensors-24-01933-f005:**
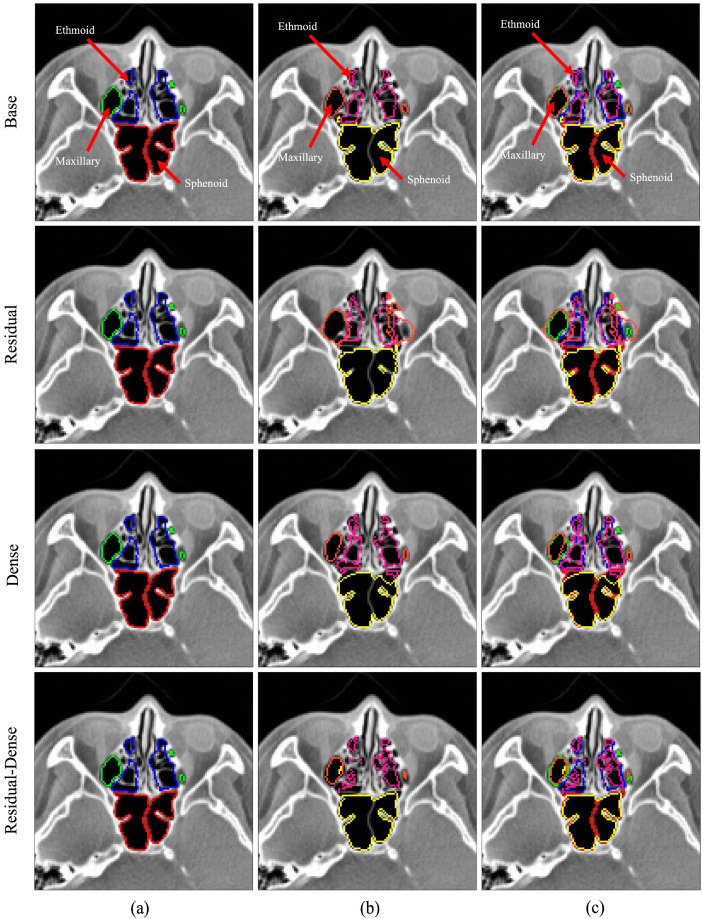
3D U-Net segmentation comparison for the normal test case. Color legend: green, orange—maxillary sinus; blue, pink—ethmoid sinus; red, yellow—sphenoid sinus; (**a**) ground truth data, (**b**) model prediction, (**c**) overlay comparison of ground truth and prediction.

**Figure 6 sensors-24-01933-f006:**
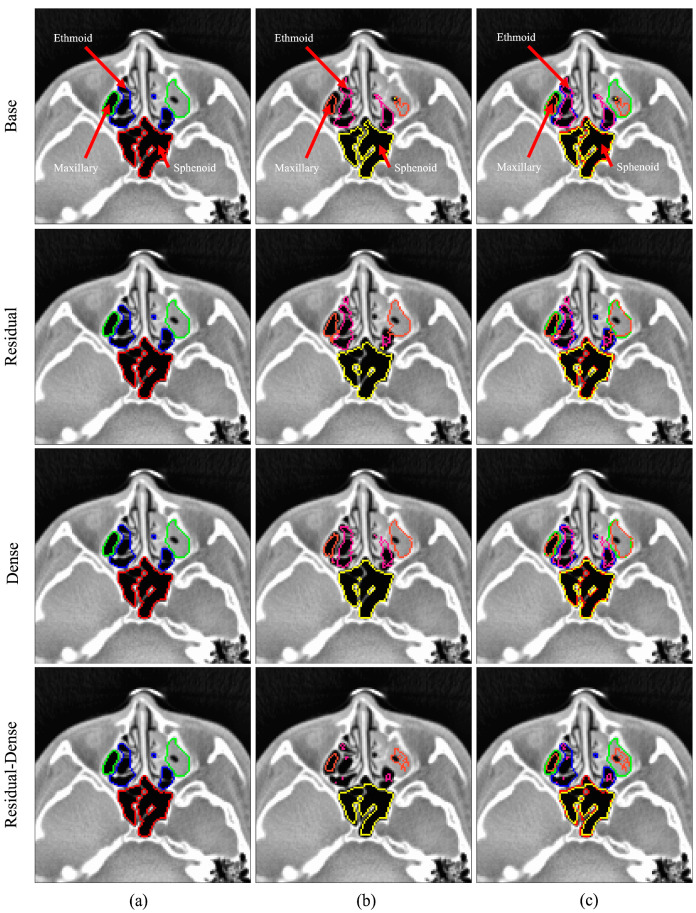
3D U-Net segmentation comparison for the abnormal test case. Color legend: green, orange—maxillary sinus; blue, pink—ethmoid sinus; red, yellow—sphenoid sinus; (**a**) ground truth data, (**b**) model prediction, (**c**) overlay comparison of ground truth and prediction.

**Figure 7 sensors-24-01933-f007:**
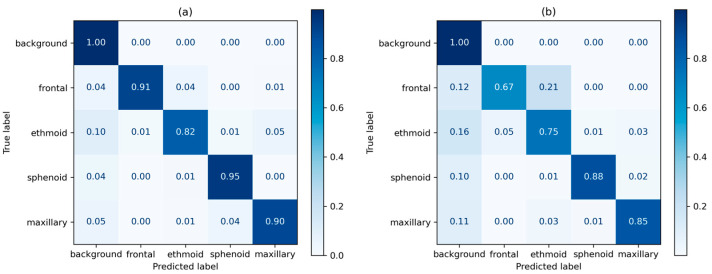
3D U-Net prediction heatmap; (**a**) prediction results for the normal test set, (**b**) prediction results for the abnormal test set.

**Table 1 sensors-24-01933-t001:** Patient distribution by age group and gender.

Age Group	Male	Gender Ratio	Female	Gender Ratio	Total	Ratio by Age
10–20	6	40.00%	9	60.00%	15	7.58%
20–30	15	68.18%	7	31.82%	22	11.11%
30–40	21	84.00%	4	16.00%	25	12.63%
40–50	14	63.64%	8	36.36%	22	11.11%
50–60	34	62.96%	20	37.04%	54	27.27%
60–70	32	71.11%	13	28.89%	45	22.73%
70–80	8	72.73%	3	27.27%	11	5.56%
80–	2	50.00%	2	50.00%	4	2.02%
Total	132	66.67%	66	33.33%	198	100%

**Table 2 sensors-24-01933-t002:** Parameter and layer count by model.

	3D U-Net		Residual		Dense		Residual-Dense	
Count	Parameter	Layer	Parameter	Layer	Parameter	Layer	Parameter	Layer
block 1	7376	4	4456	13	42,352	10	84,480	17
block 2	41,536	4	19,840	13	125,088	9	388,416	17
block 3	166,016	4	78,592	13	499,008	9	1,550,976	17
block 4	663,808	4	312,832	13	1,993,344	9	6,198,528	17
block 5	2,654,720	3	1,248,256	12	2,657,664	8	24,783,360	16
block 6	1,589,632	4	517,632	13	4,117,504	9	10,622,208	18
block 7	397,504	4	129,792	13	1,063,424	9	2,656,896	18
block 8	99,424	4	32,640	13	266,496	9	664,896	18
block 9	24,880	4	8256	13	73,872	9	138,624	18
Output	85	1	165	1	85	1	4325	1
Total	5,644,981	36	2,352,461	117	10,838,837	82	47,092,709	157

**Table 3 sensors-24-01933-t003:** Layer-by-layer kernel-wise details of each model. The 3D U-Net and dense 3D U-Net models share feature map details, while the residual 3D U-Net and residual-dense 3D U-Net models also share feature map details.

		3D U-Net/Dense 3D U-Net		Residual 3D U-Net/Residual-Dense 3D U-Net	
	Name	Feat Maps (Input)	Feat Maps (Output)	Feat Maps (Input)	Feat Maps (Output)
Encoding path	conv3d_block_1	192 × 128 × 128 × 1	192 × 128 × 128 × 16	192 × 128 × 128 × 1	192 × 128 × 128 × 32
	maxpool3d_1	192 × 128 × 128 × 16	96 × 64 × 64 × 16	192 × 128 × 128 × 32	96 × 64 × 64 × 32
	conv3d_block_2	96 × 64 × 64 × 16	96 × 64 × 64 × 32	96 × 64 × 64 × 32	96 × 64 × 64 × 64
	maxpool3d_2	96 × 64 × 64 × 32	48 × 32 × 32 × 32	96 × 64 × 64 × 64	48 × 32 × 32 × 64
	conv3d_block_3	48 × 32 × 32 × 32	48 × 32 × 32 × 64	48 × 32 × 32 × 64	48 × 32 × 32 × 128
	maxpool3d_3	48 × 32 × 32 × 64	24 × 16 × 16 × 64	48 × 32 × 32 × 128	24 × 16 × 16 × 128
	conv3d_block_4	24 × 16 × 16 × 64	24 × 16 × 16 × 128	24 × 16 × 16 × 128	24 × 16 × 16 × 256
	maxpool3d_4	24 × 16 × 16 × 128	12 × 8 × 8 × 128	24 × 16 × 16 × 256	12 × 8 × 8 × 256
Bridge		12 × 8 × 8 × 128	12 × 8 × 8 × 256	12 × 8 × 8 × 256	12 × 8 × 8 × 512
Decoding path	conv3d_trans_1	12 × 8 × 8 × 256	24 × 16 × 16 × 128	12 × 8 × 8 × 512	24 × 16 × 16 × 256
	conv3d_block_5	24 × 16 × 16 × 128	24 × 16 × 16 × 128	24 × 16 × 16 × 256	24 × 16 × 16 × 256
	conv3d_trans_2	24 × 16 × 16 × 128	48 × 32 × 32 × 64	24 × 16 × 16 × 256	48 × 32 × 32 × 128
	conv3d_block_6	48 × 32 × 32 × 64	48 × 32 × 32 × 64	48 × 32 × 32 × 128	48 × 32 × 32 × 128
	conv3d_trans_3	48 × 32 × 32 × 64	96 × 64 × 64 × 32	48 × 32 × 32 × 128	96 × 64 × 64 × 64
	conv3d_block_7	96 × 64 × 64 × 32	96 × 64 × 64 × 32	96 × 64 × 64 × 64	96 × 64 × 64 × 64
	conv3d_trans_4	96 × 64 × 64 × 32	192 × 128 × 128 × 16	96 × 64 × 64 × 64	192 × 128 × 128 × 32
	conv3d_block_8	192 × 128 × 128 × 16	192 × 128 × 128 × 5	192 × 128 × 128 × 32	192 × 128 × 128 × 5

**Table 4 sensors-24-01933-t004:** Prediction results obtained on the normal test set, reported in performance metrics per model.

Metrics	Base	Residual	Dense	Residual-Dense
F1 score	**0.843 ± 0.699**	0.785 ± 0.066	0.790 ± 0.073	0.802 ± 0.093
Accuracy	**0.995 ± 0.003**	0.992 ± 0.001	0.993 ± 0.002	0.993 ± 0.003
Precision	**0.857 ± 0.056**	0.789 ± 0.059	0.801 ± 0.060	0.822 ± 0.073
Recall	**0.854 ± 0.064**	0.821 ± 0.060	0.822 ± 0.068	0.836 ± 0.078
Mean IoU	**0.787 ± 0.071**	0.703 ± 0.067	0.714 ± 0.074	0.742 ± 0.092

**Table 5 sensors-24-01933-t005:** Prediction results obtained on the abnormal test set, reported in performance metrics per model.

Metrics	Base	Residual	Dense	Residual-Dense
F1 score	**0.793 ± 0.063**	0.741 ± 0.069	0.747 ± 0.074	0.740 ± 0.095
Accuracy	**0.994 ± 0.002**	0.991 ± 0.002	0.992 ± 0.002	0.991 ± 0.003
Precision	**0.839 ± 0.057**	0.779 ± 0.067	0.785 ± 0.071	0.793 ± 0.089
Recall	**0.785 ± 0.067**	0.755 ± 0.076	0.756 ± 0.068	0.745 ± 0.092
Mean IoU	**0.717 ± 0.061**	0.653 ± 0.063	0.666 ± 0.074	0.670 ± 0.089

## Data Availability

The raw data supporting the conclusions of this article will be made available by the authors on request.

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
