# Peer review of "Deep Learning-Based Multi-Class Segmentation of the Paranasal Sinuses of Sinusitis Patients Based on Computed Tomographic Images"

_sensors, 2024, doi:10.3390/s24061933_

Round 1

Reviewer 1 Report

Comments and Suggestions for Authors

Summary: The article presents a comprehensive study on the development and evaluation of a multiclass Convolutional Neural Network (CNN) segmentation model for paranasal sinus segmentation in CT images, with a focus on comparing four variations of the 3D U-Net model. 

Strength:

1. The paper addresses a significant problem in the medical field, aiming to improve the accuracy of paranasal sinus segmentation, which is crucial for surgical planning and guidance. The comparison of different 3D U-Net variations (normal, residual, dense, and residual-dense) is a commendable approach, contributing to the existing body of knowledge on the subject.

2. The methodology is well-structured, with a clear explanation of the data collection, pre-processing, and model training phases. The presentation of results is clear, with performance metrics such as IOU, accuracy, recall, precision, and F1-score thoroughly evaluated.

Limitations:

1. While the current study's 5-class segmentation model demonstrates substantial merit, it presents a slightly lower performance metric compared to Kuo's 6-class segmentation model. It would be beneficial for the authors to elucidate the unique contributions and potential advantages of their model in the broader context of paranasal sinus segmentation. Understanding the specific clinical scenarios or particular challenges where this model could offer superior utility would greatly enhance the value of the study.

2. A more detailed analysis of the features derived from the CT scans could significantly enrich the study. Specifically, identifying and discussing the features most correlated with accurate segmentation, particularly in cases presenting mucosal inflammation, could provide critical insights. Such an analysis might not only enhance the current model's performance but also contribute to the broader academic discourse on CT image segmentation.

3. On line 52, the reported consistency of the DSC value before and after post-processing could benefit from clarification. It would be helpful to either elaborate on the reasons behind this consistency or present the data with more significant figures to portray a more nuanced view of the post-processing impact.

4. The paper could provide further clarity on the distribution of the training and validation datasets in relation to the test dataset. Ensuring and demonstrating that the training, validation, and test datasets are representative and well-distributed can significantly strengthen the validity of the model's evaluation and the overall findings. A more detailed discussion or visual representation of the dataset distributions might be beneficial in solidifying the study's robustness.

Author Response

  1. While the current study's 5-class segmentation model demonstrates substantial merit, it presents a slightly lower performance metric compared to Kuo's 6-class segmentation model. It would be beneficial for the authors to elucidate the unique contributions and potential advantages of their model in the broader context of paranasal sinus segmentation. Understanding the specific clinical scenarios or particular challenges where this model could offer superior utility would greatly enhance the value of the study.

Response: One of the advantages of our approach is its ability to segment the sinus volumes of sinusitis patients with mucosal inflammation. Our model suffered from lower performance metrics due to difficulty of identifying sinus cavities with mucosal inflammation. We believe that our approach offers better flexibility in terms of clinical applications as we expect our model to

  1. A more detailed analysis of the features derived from the CT scans could significantly enrich the study. Specifically, identifying and discussing the features most correlated with accurate segmentation, particularly in cases presenting mucosal inflammation, could provide critical insights. Such an analysis might not only enhance the current model's performance but also contribute to the broader academic discourse on CT image segmentation.

Response: The investigation into which feature contributes most to the accurate segmentation of the sinus is currently ongoing. We have incorporated several sentences in lines 248-249, page 10, outlining our hypothesis regarding why sinus cavities are more challenging to segment in cases presenting mucosal inflammation. Furthermore, we suggest that achieving better contrast between the sinus bone and cavity could be crucial for the accurate segmentation of the sinus.

  1. On line 52, the reported consistency of the DSC value before and after post-processing could benefit from clarification. It would be helpful to either elaborate on the reasons behind this consistency or present the data with more significant figures to portray a more nuanced view of the post-processing impact.

Response: Based on the study conducted by Choi et al., titled "Deep learning-based fully automatic segmentation of the maxillary sinus on cone-beam computed tomographic images," the post-processing technique included filtering out isolated volumes that did not meet specific size criteria. While the complete advantage of this post-processing technique may not be entirely clear from our perspective, we opted not to adopt a similar approach. This decision was made because our methodology also entails segmenting various volumes of the sinus, and removing such isolated volumes could potentially impair the performance of our models.

  1. The paper could provide further clarity on the distribution of the training and validation datasets in relation to the test dataset. Ensuring and demonstrating that the training, validation, and test datasets are representative and well-distributed can significantly strengthen the validity of the model's evaluation and the overall findings. A more detailed discussion or visual representation of the dataset distributions might be beneficial in solidifying the study's robustness.

Response: We have added a summary of the data distribution in lines 82-83 to provide a clearer delineation of our dataset situation.

Reviewer 2 Report

Comments and Suggestions for Authors

  • Dear authors, I am pleased to offer my genuine suggestion, hoping it may contribute to your publication.
    • While deep learning is a prominent aspect mentioned in the title and keywords of the article, the method section lacks detailed elaboration, consisting of only two paragraphs. Considering not all readers are experts in deep learning, addressing the following questions could enhance the logical clarity and robustness of the research: What challenges did the authors face in applying such methods to the current topic, and what solutions were implemented? How does the U-Net variation cited by the authors differ from others, and what improvements have been made? Regarding the results, is there an explanation for how one U-Net variation outperforms others?
    • Although the authors conducted various experiments comparing different U-Net structures, it would strengthen the paper to include at least one diagram and one table detailing the models' architecture, deep learning units, number of parameters, depth, and other relevant information. This would allow readers understand the differences between models and how they correlate with the final performance in the shortest time.
    • Please describe more about training tricks, such as training epoch, early stopper, learning rate plateaus, etc., to facilitate result replication. If similar methods from other works are applied, specify how key parameters are determined.
    • Other than Figure 4 and 5, it is suggested that other two points of view (as in Figure 1) shall be given.
  • Minor revisions are suggested:
    • For the dataset, I suggest to include more details like age, gender, ethnicity, or provide the citation/web-link if there is a specific paper discussing it. As mentioned in the paper by the authors, additional information aids the research community in evaluating the method's generalizing power more effectively.
    • A few more sentences explaining why and how the window-based method should be applied will make data preprocessing part more comprehensive in logic.

Author Response

What challenges did the authors face in applying such methods to the current topic, and what solutions were implemented?

Response: One of the challenges we faced while applying our model in the segmentation of the sinuses, was with the volumetric ratio of our data. As CT scans can have varying slice thickness depending on the operator and various circumstances, the pixel-wise reconstructed volume from these scans can have disproportionate ratios. To overcome this, we used an isotropic voxel reconstruction algorithm outlined in our method section.

How does the U-Net variation cited by the authors differ from others, and what improvements have been made?

Response: Each variation of the U-Net architecture differs by incorporating an additional architectural layer as indicated in its name. For instance, the Residual 3D U-Net substitutes the original convolution blocks with a blend of convolution and residual blocks.

Regarding the results, is there an explanation for how one U-Net variation outperforms others?

Response: While we lack an exact scientific explanation for why the 3D U-Net outperformed other architectures, we suspect that the additional depth and complexity introduced by the other models may have led to overfitting. However, as these are speculative observations, we have refrained from including such explanations in our paper to maintain its objectivity.

Include at least one diagram and one table detailing the models' architecture, deep learning units, number of parameters, depth, and other relevant information.

Response: We have included a structural diagram of the 3D U-Net alongside a table detailing the parameter and layer count for each model.

Please describe more about training tricks, such as training epoch, early stopper, learning rate plateaus, etc., to facilitate result replication.

Response: We have added details regarding the training configuration in page 4, line 152. “The tolerance for learning rate reduction was configured to 20 epochs, while the early stopper tolerance was set at 30 epochs”.

Minor revisions are suggested:

For the dataset, I suggest including more details like age, gender, ethnicity, or provide the citation/web-link if there is a specific paper discussing it. As mentioned in the paper by the authors, additional information aids the research community in evaluating the method's generalizing power more effectively.

Response: We included a table presenting the distribution of patient age and gender. Given that all data collected for this study originated from East Asians, we deemed it unnecessary to include information regarding the ethnicity of the patients.

A few more sentences explaining why and how the window-based method should be applied will make data preprocessing part more comprehensive in logic.

Response: We chose to incorporate window setting into our preprocessing algorithm as it affects the contrast settings of the CT images. The specific methodology for setting the window was performed by the radiologist operating the machine and we consider this information out of scope for the topic of this paper.

Round 2

Reviewer 2 Report

Comments and Suggestions for Authors

Dear authors,

Glad to see the revision, and it is much elaborative in method details to me. Only one minor part is, to figure 3 or table 2, please add kernel-wise details for different layers (e.g., 3x3x1 3D CNN, or 2x2x1 3D MaxPooling etc.). Looking forward to your update.

https://miro.medium.com/max/3120/1*aRMefObpm7AMVOZYYiQAMQ.png

https://www.researchgate.net/publication/329041317/figure/tbl1/AS:694689313353729@1542638247116/Layer-placement-of-the-proposed-hyper-dense-connected-UNet.png

Author Response

Only one minor part is, to figure 3 or table 2, please add kernel-wise details for different layers (e.g., 3x3x1 3D CNN, or 2x2x1 3D MaxPooling etc.). Looking forward to your update.

Response: Thank you for your suggestion. We have incorporated kernel-wise details for each layer in a separate table (Table 3). We decided against adding this additional information to Table 2 to prevent it from becoming excessively large.